# AMPK Amplifies IL2–STAT5 Signaling to Maintain Stability of Regulatory T Cells in Aged Mice

**DOI:** 10.3390/ijms232012384

**Published:** 2022-10-16

**Authors:** Ram Hari Pokhrel, Ben Kang, Maheshwor Timilshina, Jae-Hoon Chang

**Affiliations:** 1College of Pharmacy, Yeungnam University, Gyeongsan 38541, Korea; 2Department of Pediatrics, School of Medicine, Kyungpook National University, Daegu 41944, Korea

**Keywords:** aging, AMPK, inflammation, Treg stability

## Abstract

AMP-activated protein kinase (AMPK), an important regulator of the aging process, is expressed in various immune cells. However, its role in regulatory T cell (Treg) stability during aging is poorly understood. Here, we addressed the role of AMPK in Treg function and stability during aging by generating Treg-specific AMPKα1 knockout mice. In this study, we found that AMPKα1-deficient Tregs failed to control inflammation as effectively as normal Tregs did during aging. AMPK knockout from Tregs reduces STAT5 phosphorylation in response to interleukin (IL)-2 stimulation, thereby destabilizing Tregs by decreasing CD25 expression. Thus, our study addressed the role of AMPK in Tregs in sensing IL-2 signaling to amplify STAT5 phosphorylation, which, in turn, supports Treg stability by maintaining CD25 expression and controlling inflamm-aging.

## 1. Introduction

Regulatory T cells (Tregs) play a critical role in maintaining immune tolerance to self-antigens and suppressing excessive immune responses that may cause collateral damage to the host [1]. However, only stable Tregs play a regulatory role [2]. The transcription factor Foxp3 and the α-subunit of interleukin (IL)-2 receptor CD25 are required for the stability and suppressive function of Tregs [3]. However, in conditions such as aging, a proportion of Tregs display altered stability and cause loss of Treg function, rendering the host susceptible to chronic-smoldering inflammation [4]. Some Treg populations, mainly naturally occurring Tregs (nTregs), accumulate with advancing age, whereas induced Tregs (iTregs) are less available in elderly hosts [4].

AMP-activated protein kinase (AMPK) is a highly conserved serine/threonine protein kinase that acts as a master metabolism regulator and maintains energy homeostasis during metabolic stress, both at the cellular and physiological levels [5,6,7]. Besides the well-established characterized role of AMPK in regulating nutrient metabolism, it is expressed in various immune cells and an important regulator of the aging process [8,9].

Inflammation is one of the seven evolutionarily conserved mechanistic pillars of aging shared by age-related diseases [10]. The immune system is susceptible to age-related changes [4]. Alterations in lymphocyte subsets and cytokine dysregulation play vital roles in the inability to control systemic inflammation and seem to be markers of unsuccessful aging [11,12,13]. The causes of “inflamm-aging” are not clearly understood. Many recent studies have established a role for AMPK in inflammation; however, the relationship between AMPK in regulatory T cells and inflammation during aging remains unclear.

In this study, we addressed the role of AMPK in Treg stability and function during aging by generating Treg-specific AMPKα1 knockout mice. We found that AMPK maintained the stability of Tregs and played a vital role in decreasing “inflamm-aging.”

## 2. Results

### 2.1. Young AMPKα1^fl/fl^Foxp3^yfp-cre^ (AMPK-KO) and Foxp3^ypf-cre^ (WT) Mice Are Phenotypically Similar

To address the specific role of AMPK in Treg stability, *AMPKα1^fl/fl^* mice were crossed with *Foxp3^YFP-Cre^* mice to generate *AMPKα1^fl/fl^Foxp3^YFP-Cre^* mice with Foxp3-dependent specific AMPKα1 deletion (AMPK-KO) and *Foxp3^yfp-Cre^* mice as wild-type controls (WT). Generating mice with Treg-specific AMPK deletion was confirmed by Western blotting, as AMPK expression is substantially reduced in Tregs from AMPK-KO mice (Appendix A). We have previously showed that AMPK-KO and WT mice were phenotypically similar and the number and percentage of CD4^+^ and CD8^+^ T cells, Th1, Th17, and Tregs in the spleen, mesenteric lymph nodes (MLN), and peripheral lymph nodes (PLN) were not significantly different [14]. In addition, we compared the other phenotypes between the two mice groups. We checked the mRNA expression of *Tbet*, *Gata3*, *Rorγt*, and *Foxp3*, which are the transcription factors Th1, Th2, Th17, and Tregs, respectively, normalized to *Gapdh* mRNA expression, and found that they were not significantly different between the two mice groups (Figure 1A). We then analyzed whether there were any differences in naive CD4^+^ T cells; however, the percentage of CD62L was comparable (Figure 1B). Tregs are a subset of CD4^+^ T cells; therefore, we checked the surface markers for CD4^+^ T cells and found that the CD69, CD39, CD132, CD122, CD28, CXCR5, CCR1, and CCR6 expression levels were comparable between the two phenotypes (Figure 1C). Since AMPK was conditionally deleted from Tregs, we analyzed the thymus to determine whether there were any changes in Tregs in the developmental phase, but there was no difference in the percentage of immature and mature thymic Tregs (Figure 1D). In addition, in the peripheral region, the helios in Tregs, which is a marker for thymic Tregs, were similar between WT and AMPK-KO mice (Figure 1E). Similarly, the percentage of CD25 gated from CD4^+^YFP^+^ cells were comparable in the thymus, MLN, PLN, and spleen of WT and AMPK-KO mice (Figure 1F). Tregs express various surface markers, such as CTLA-4, GITR, OX40, CD73, CD39, Nrp1, ICOS, and PD-1. The expression of CTLA-4, GITR, OX40, CD73, and CD39 were comparable in WT and AMPK-KO mice; however, Nrp1, ICOS, and PD-1 expression levels increased in AMPK-KO mice [14]. Consistent with previous findings, the expression of inhibitory receptor PD-1 increased in AMPK-deleted Tregs in this study (Figure 1G). In addition, the percentage of germinal center B cells (Appendix A) and follicular helper T cells (Appendix A), and serum IgG, IgG1, IgG2a, IgG2b, IgG3, IgE, and IgM (Appendix A) concentrations, were comparable between WT and AMPK-KO mice, indicating no alterations in humoral immunity. Thus, these findings suggest that AMPK loss in Tregs caused no prominent phenotypic differences in C57BL/6J mice.

### 2.2. Young AMPK-KO Mice Showed a Less Inflammatory Phenotype Than WT Mice

Several studies have described how AMPK activation can inhibit the inflammatory response, whereas decreased AMPK activity is associated with increased inflammation [15,16]. To determine the role of AMPK in Tregs in inflammation in young mice, we used the experimental autoimmune encephalomyelitis (EAE) model, a well-established mouse model for multiple sclerosis. Accordingly, EAE was induced in mice by immunization with the MOG_35–55_ peptide in CFA and pertussis toxin, as explained in Section 4.2. Interestingly, young AMPK-KO mice showed low disease severity, as indicated by decreased clinical scores (Figure 2A). Furthermore, we isolated the brain and spinal cord from EAE-induced WT and AMPK-KO mice and analyzed CD4^+^ and CD8^+^ T cell infiltration to evaluate immune cell infiltration in nervous tissues. CD4^+^ T and CD8^+^ T cells infiltrated the spinal cord of WT mice more than the spinal cord of AMPK-KO mice (Figure 2B,C), but not the brain tissues (Figure 2B,C). Previous studies have reported that Th1 and Th17 cells are critically involved in the pathogenesis of multiple sclerosis and its mouse model, EAE [17,18,19]. Therefore, we examined TH1 and TH17 cell infiltration by intracellular staining of mononuclear cells derived from the spinal cord and lymph nodes. Supporting the previous result, IFN-γ-specific Th1 cells significantly reduced in the spinal cord and lymph nodes of AMPK-KO mice (Figure 2D). Collectively, these findings indicate that AMPK-KO mice show less EAE than WT mice by reducing T cell infiltration in the spinal cord.

### 2.3. AMPK Loss Abolishes the Ability of Tregs to Resolve EAE in Aged Mice

As we observed a less inflammatory phenotype in AMPK-KO mice, we then checked whether the aged mice (24–32 weeks) showed similar results. However, contrary to young mice, aged mice showed the opposite results. Monitoring EAE-induced mice revealed that aged WT mice showed less disease severity than AMPK-KO mice, as indicated by diminished clinical scores (Figure 3A). Additionally, the spinal cord was isolated and homogenized, and more mononuclear cellularity was observed in the AMPK-KO mice than that in the WT mice (Figure 3B). In addition, both WT and AMPK-KO mice showed substantial infiltration of CD4^+^ and CD8^+^ T cells. However, CD4^+^ and CD8^+^ T cell infiltration increased in the spinal cord and brain of AMPK-KO mice (Figure 3C). Pro-inflammatory cytokines play essential roles in EAE progression; therefore, we checked the presence of pro-inflammatory cytokines IFN-γ in CD4^+^ and CD8^+^ T cells and IL-17A in CD4^+^ T cells by intracellular staining of mononuclear cells derived from the spinal cord, brain, and spleen. IFN-γ^+^ CD4^+^ and IFN-γ^+^ CD8^+^ T cells increased in the AMPK-KO mice’s spinal cords (Figure 3D,E), but no significant difference was observed in IFN-γ^+^ CD4^+^ T cells in the spleen (Figure 3F) and IL-17A^+^ CD4^+^ T cells in both the spinal cord and spleen (Figure 3E,F) of the AMPK-KO mice than those in their WT counterparts. Moreover, histological analysis of the spinal cord revealed increased cellular infiltration in the histological sections of the AMPK-KO mice spinal cord (Figure 3G). Altogether, these findings revealed that AMPK-KO aged mice showed more severe EAE than the WT aged mice due to higher infiltration of T cells and pro-inflammatory cytokine IFN-γ in the spinal cord.

### 2.4. Treg-Specific AMPK Deletion Develops Spontaneous Systemic Lymphoproliferative Disease with Age

In AMPK-KO mice, EAE progression was more severe; therefore, we analyzed the phenotypes of aged WT and AMPK-KO mice to determine the phenotypic differences between these two groups of mice. We first examined the size of secondary lymphoid organs and found that AMPK-KO mice had a comparatively larger spleen and PLN (Figure 4A), as well as higher cellularity (Figure 4B) than WT mice. We then analyzed CD4^+^ T cells for the expression of homing receptors CD44 and CD62L to distinguish naive (CD44^lo^ CD62L^hi^) and effector memory (CD44^hi^ CD62L^lo^) cells in the spleen, MLN, and PLN. Notably, the effector memory CD4^+^ T cells significantly increased in AMPK-KO mice (Figure 4C). These results suggest the development of an inflammatory phenotype in AMPK-KO mice. Therefore, we further investigated the inflammatory cytokines IFN-γ in CD4^+^ and CD8^+^ T cells and IL-17A in CD4^+^ T cells by isolating cells from the spleen, MLN, and PLN. Although the percentage of CD4^+^IL-17A^+^ cells was comparable between WT and AMPK-KO mice (Figure 4D), the percentage of CD4^+^IFN-γ^+^ and CD8^+^IFN-γ^+^ cells significantly increased in AMPK-KO mice (Figure 4D,E). Moreover, serum IgG1, IgE, and antibodies to double-stranded DNA concentrations were elevated, whereas the IgG and IgM levels were unchanged in AMPK-KO mice (Appendix A). Altogether, these results suggest that AMPK-KO mice develop a more inflammatory phenotype over time than their WT counterparts.

### 2.5. Substantial Decrease of CD25 Expression in AMPK-KO Tregs with Age

The RNA of Tregs from aged WT and AMPK-KO mice were sequenced. GSEA plot analysis showed decreased expression of IL-2-responsive Foxp3 target genes (Appendix A). In addition, both the GSEA plots (Appendix A) and heat maps (Appendix A) showed enriched inflammation-related genes in AMPK-KO mice, signifying that aged AMPK-KO Tregs lost their function. Therefore, we examined the percentage of Tregs due to the decreased function of AMPK-KO Tregs and lymphoproliferative disease development in AMPK-KO mice. Accordingly, we analyzed the Foxp3 percentage in the spleen, PLN, MLN, and thymus; interestingly, AMPK-KO mice showed more CD4^+^Foxp3^+^ cells in spleen, PLN, and MLN than WT mice (Figure 5A). In addition, Treg proliferation increased in AMPK-KO mice (Figure 5B), suggesting that the development of the inflammatory phenotype in aged AMPK-KO mice was not due to Treg reduction. Therefore, we hypothesized that the inflammatory phenotype might be due to Treg instability. CD25 is required for Treg development, peripheral maintenance, and suppression [20]. Thus, we assessed CD25^+^Foxp3^+^ and CD25^−^Foxp3^+^ Tregs and found that the CD25^−^Foxp3^+^ population increased in the spleen, MLN, and PLN (Figure 5C), and CD25 expression was significantly reduced in the thymus, spleen, MLN, and PLN in AMPK-KO Tregs (Figure 5D). To evaluate the stability of AMPK-deficient Tregs, we sorted Foxp3^+^CD25^+^Tregs from both WT and AMPK-KO mice and cultured the cells in the presence of IL-2 time-dependently. Remarkably, CD25 expression decreased in AMPK-KO Tregs by 18 h and further diminished at 2 and 6 d (Figure 5E). Since the Foxp3^+^CD25^−^ population increased in AMPK-KO Tregs, we performed the same experiment using WT Tregs in the presence of compound C. To our expectation, compound C reduced CD25 expression time-dependently (Figure 5F), which proved that AMPK-deficient Tregs downregulated CD25 expression time-dependently. Overall, we can conclude that aged AMPK-KO Tregs lose their stability by CD25 expression loss.

### 2.6. AMPK Supports Suppressive Activity of Tregs

To further examine whether AMPK is dispensable for Tregs to inhibit inflammation, we analyzed the ability of AMPK-deficient Tregs from aged mice to suppress CD45RB^hi^ cell transfer-induced colitis in Rag^−/−^ mice. Disease onset, such as body weight loss, was detected 5 weeks post-CD45RB^hi^ cell transfer. However, co-transfer of AMPK-deficient Tregs failed to increase the body weight of colitis-induced mice as effectively as WT Tregs (Figure 6A). In addition, morphological changes were observed in the colon, spleen, and MLN. Compared to that in WT Treg-recipient mice, the colons shortened, and spleens and MLNs enlarged in AMPK-KO Treg-recipient mice (Figure 6B,C). Furthermore, histological analysis of the colon showed more inflammatory infiltrate, disrupted mucosa, and crypt architecture in AMPK-KO Treg-recipient mice than those in the WT Treg-recipient mice (Figure 6D). Finally, we tested CD25 expression in Tregs from the spleen and MLN and found that Tregs of both the spleen and MLN of mice co-transferred with AMPK-KO Tregs showed less CD25 expression (Figure 6E). Altogether, mice with T cell transfer colitis showed that AMPK-KO Tregs have less suppressive ability than WT Tregs, possibly because of the decreased CD25 expression.

### 2.7. AMPK Amplifies IL-2–STAT5 Signaling in Tregs

Previous studies have suggested that activated STAT5 translocates into the nucleus of cells to regulate target genes, including CD25 [21,22,23]. Therefore, we examined the expression of STAT5 and other STAT proteins by Western blotting. Western blot analysis revealed that the expression of phosphorylated STAT5 was reduced in Tregs of AMPK-KO mice after TCR and IL2 stimulation (Figure 7A). We also analyzed other STAT proteins, such as STAT3 (Figure 7B) and STAT4 (Figure 7C), but their expression levels were comparable. Moreover, we analyzed STAT5 expression in Tregs of aged mice and observed that p-STAT5 was greatly reduced in the Tregs of AMPK-KO mice (Figure 7D). Thereafter, we treated WT Tregs with compound C along with TCR and IL-2 stimulation for 18 h and found that the AMPK inhibitor compound C decreased p-STAT5 expression (Figure 7E). Overall, the data indicate the role of AMPK in STAT5 phosphorylation and CD25 maintenance in Tregs.

## 3. Discussion

Although the roles of Tregs have been established, the mechanism through which AMPK maintains Treg stability during aging remains poorly understood. In this study, we demonstrate the crucial role of AMPK as a driving force for IL-2–STAT5-mediated Treg stability and suppressive function. AMPK loss in Tregs weakens IL-2–STAT5 signaling, thereby reducing CD25 expression; thus, AMPK-deficient Tregs fail to control inflammation during aging compared to normal Tregs. Thus, our data highlight the role of AMPK in Tregs in maintaining the IL-2–STAT5 axis for the effective suppressive function of Tregs during aging.

Similar to other organ systems in the body, the immune system is susceptible to age-related changes. Recent studies have suggested that Tregs are characterized by the presence of the transcription factor Foxp3, and CD25 expression plays a critical role in regulating immune responses and controlling inflammation [4,24,25].

*Tbet, Gata3, Rorγt,* and *Foxp3* expression levels as well as the expression of CD4^+^ T cell surface markers such as CD69, CD39, CD132, CD122, CD28, CXCR5, CCR1, CCR6 were not significantly different. CD40L is a co-stimulatory receptor on Tfh cells [26] and also helps in immunoglobulin class switching, but there is no significant difference between GCB, Tfh cells, and concentrations of IgM and other immunoglobulins, indicating that expression of CD40L is similar in young WT and AMPK-KO mice. We have previously showed similar expression of Treg surface markers, except PD-1, Nrp1, and ICOS [14], suggesting that the newly generated young WT and AMPK-KO mice are almost phenotypically similar and have similar Th1, Th2, Th17, and Treg transcriptional profiles. This showed that the inflammatory phenotype observed in aged AMPK-KO mice developed over time, but not during birth and early age.

We observed less EAE in 6–8-week-old AMPK-KO mice than that in WT mice, which is opposite to the results obtained in aged mice. We have previously showed higher PD-1 expression in Tregs of AMPK-KO mice than in those of WT mice, which is responsible for the higher suppressive activity of Tregs [14]. Moreover, P38 is required for the effector phase of EAE [27] and AMPK-KO mice showed less p-P38 expression in our previous study [14]. Therefore, the lower EAE in AMPK-KO mice might be due to less P38 expression and the higher suppressive ability of Tregs due to higher PD-1 expression.

Previous studies have suggested that aging is responsible for the inflammatory phenotype [28,29,30] and activated AMPK inhibits inflammation [31,32]. Despite the importance of AMPK in inflammation regulation, its role in Tregs during aging is unknown. Therefore, our study focused on the role of AMPK in Tregs during aging to cope with inflamm-aging, and the results suggest that AMPK in Tregs can reduce inflammation. During aging, AMPK-KO mice showed lymphoproliferative disease, with a larger spleen and lymph nodes than their WT counterparts, and more effector phenotypes, with higher production of the inflammatory cytokine IFN-γ.

The etiology of autoimmunity and lymphoproliferation in AMPK-KO mice might be multifactorial, but we provided evidence that Tregs from these mice exhibited functional instability due to increased Foxp3^+^CD25^−^ Treg population. Previous studies have suggested that CD4^+^CD25^high^Foxp3^+^ Tregs are more suppressive [33,34,35], and the inability of AMPK-KO mice to resolve EAE might be due to a decrease in suppressive capacity because of less CD25 expression in Tregs during aging.

Activated STAT5 translocates into the nucleus, which regulates the transcription of the target genes, including IL-2 receptor α (CD25) [21,22,23]; therefore, we focused on analyzing STAT5 in Tregs. Notably, STAT5 phosphorylation was reduced in IL-2-stimulated AMPK-deficient Tregs. This result provides supportive evidence for decreased CD25 expression in AMPK-KO mice. Furthermore, the AMPK inhibitor compound C decreased STAT5 phosphorylation, which reinforced our findings. However, the mechanism of AMPK-mediated STAT5 phosphorylation remains unclear, and further studies are needed to elucidate the underlying mechanism.

## 4. Materials and Methods

### 4.1. Mice

C57BL/6J, *Rag^−/−^, AMPKα1^fl/fl^,* and *Foxp3^YFP-Cre^* mice were purchased from the Jackson Laboratory (Bar Harbor, ME, USA). *AMPKα1^fl/fl^* mice were crossed with *Foxp3^YFP-Cre^* mice to generate AMPK-KO mice. The 6–8- and 24–32-week-old AMPK-KO mice were used as young and aged mice, respectively. Age- and sex-matched littermate *Foxp3^yfp-cre^* mice were used as controls. The other mice were used at 6–10 weeks of age unless otherwise stated. All mice were maintained under specific pathogen-free conditions in the animal facilities of Yeungnam University. All animal experiments were approved by the Institutional Animal Care Committee (IACUC) of Yeungnam University.

### 4.2. EAE Models

AMPK-KO, WT, and C57BL/6J mice were subcutaneously immunized by mixing synthetic peptides derived from myelin oligodendrocyte glycoprotein (MOG_35–55_) peptide (MEVGWYRSPFSRVVHLYRNGK, China Peptides, Shanghai, China) in CFA containing 10 mg/mL of a heat-killed H37Ra strain of *Mycobacterium tuberculosis* (Chondrex, Inc., Woodiville, WA, USA). Immediately and 48 h after MOG injection, the mice were intraperitoneally immunized with 250 ng pertussis toxin (Tocris Bioscience, Bristol, UK). The mice were examined daily and disease severity was recorded using a standard clinical score: 0, no obvious clinical change; 0.5, limp tail with no upright tail function; 1.5, wobbly gait; 2, flaccid tail with partial hind limb paralysis; 3, complete hind limb paralysis; 4, moribund state; and 5, dead.

### 4.3. Cell Purification and Culture

The mice were euthanized, and lymphocytes were isolated from lymphoid organs (spleen, PLNs, MLNs, and thymus). CD4^+^ T cells were purified by positive selection using CD4-conjugated magnetic beads, and Tregs were sorted using a BD FACSJazz Cell Sorter. The sorted cells were used for in vitro culture in complete medium (RPMI medium containing 10% FBS and 0.1% antibiotics). Tregs were activated with plate-coated 5 μg/mL anti-CD3, 1 μg/mL anti-CD28, and 10 ng/mL mouse recombinant IL-2. The cells were cultured in a cell incubator with 5% CO_2_ at 37 °C for 72 h.

### 4.4. Flow Cytometry and Cell Sorting

Single-cell suspensions were prepared from the PLNs, MLNs, and the spleen. The cells were then lysed with red blood cell lysis buffer (Cat# 420301, BioLegend; San Diego, CA, USA), washed with RPMI, and resuspended in complete medium (RPMI containing medium with 10% FBS and 0.1% antibiotics). For analyzing the different markers, the cells were stained with 1:100 diluted fluorescent-labeled anti-CD3 (Clone# 145-2C11), anti-CD4 (Clone# GK 1.5), anti-CD8 (Clone# SK1), anti-IL-4 (Clone# 11B11), anti-CD25 (Clone# PC61), anti-CD28 (Clone# 37.51), anti-CD45RB (Clone# RA3-6B2), anti-CD39 (Clone# Duha59), anti-CD73 (Clone# TY/11.8), anti-ICOS (Clone# 7E.17G9), anti-PD-1 (Clone# RMPI-30), anti-CD304 (Nrp1, Clone# 3E12), anti-CD357 (GITR, Clone# DTA-1), anti-CTLA-4 (Clone# UC10-4B9), anti-CD44 (Clone# IM7), anti-CD62L (Clone# MEL-14), anti-IFN-γ (Clone# XMG1.2), and anti-IL-17 (Clone# TC11-18H10.1), purchased from BioLegend. For intracellular staining, the cells were stimulated for 4–6 h with 50 ng/mL PMA (Cat# 79346, Sigma Aldrich, St. Louis, MI, USA) plus 750 ng/mL ionomycin (Cat# I9657, Sigma Aldrich) in the presence of 1× Golgistop (Cat# 51-2092KZ, BD Biosciences; Franklin Lakes, NJ, USA). Then, the cells were surface-stained with anti-CD4 and anti-CD8 antibodies, followed by fixation and permeabilization using a commercial buffer (Cat# 554714, BD Biosciences, Franklin Lakes, NJ, USA). For Foxp3 staining, cells were fixed and permeabilized using a Foxp3/Transcription Factor Staining buffer (Cat#00-5521-00, Thermo Fisher Scientific; Waltham, MA, USA). The IFN-γ^+^ Th1 cell, IL-17A^+^ Th17 cell, and Foxp3^+^ Treg counts were determined by flow cytometry. YFP^+^CD25^+^ Tregs were sorted using a BD FACSJazz™ cell sorter. Flow cytometry data were acquired using a BD FACSVerse flow cytometer and analyzed using FlowJo software, version 10.2 (FlowJo LLC; Ashland, OR, USA).

### 4.5. Total RNA Isolation and Quantitative Real-Time PCR

RNA was extracted from sorted YFP^+^CD25^+^ Tregs using the ReliPrep ™ RNA Cell Miniprep System (Cat #Z6011, Promega Corporation; Madison, WI, USA). cDNA was synthesized using the Goscript Reverse Transcription System (Cat#A5001; Promega Corporation, Madison, USA). Using the QuantiTect SYBR Green PCR kit, the mRNA expression level of each gene was measured using a real-time PCR system. Real-time PCR was done following thermal cycle plan: 60 min at 42 °C (for reverse transcription), 15 min at 95 °C (for heat inactivation or pre-denaturation), and 40 cycles for 15 s at 95 °C, 30 s at 58 °C, and 30 s at 72 °C. All data were analyzed using the comparative C_T_ method and the fold change was calculated as previously described [14,36].
2^−^^△△C(T)^ = [(C_T_ of gene of interest − C_T_ of internal control) Sample A − (C_T_ of gene of interest − C_T_ of internal control) Sample B)]

Melting curve analysis was performed to check for non-specific amplification and confirm that a single amplicon was generated by qPCR. The PCR efficiency was >90%. PCR target genes and primer sequences are listed in Appendix A.

### 4.6. T Cell Transfer Colitis

For naive T cell transfer colitis, 4 × 10^5^ CD4^+^CD45RB^hi^CD25^−^ T cells from B6.SJL female mice expressing the CD45.1 congenic marker and 2 × 10^5^ YFP^+^ Tregs from female WT or AMPK-KO mice were prepared by flow cytometry. *Rag1^−/−^* female mice were intraperitoneally injected with 4 × 10^5^ CD4^+^CD25^−^CD45RB^hi^ T cells, either alone or in combination with 2 × 10^5^ Tregs from WT or AMPK-KO mice. The mice were assessed weekly for clinical signs of colitis and analyzed 12 weeks after the transfer. Lymphocytes were isolated from the spleen and mesenteric lymph nodes and analyzed using flow cytometry.

### 4.7. Histopathology

The spinal cords and colons were removed from the mice. The samples were fixed in formalin and processed for sucrose embedding and cutting using a sliding microtome (Microm HM 450, ThermoFisher Scientific, Walldorf, Germany). Sections (20 μM) were cut, stained with hematoxylin and eosin (H&E), and examined under a light microscope.

### 4.8. Enzyme-Linked Immunosorbent Assay (ELISA)

Serum was collected from 6–8- and 24–30-week-old AMPK-KO and WT mice via retro orbital puncture for analyzing the Ig antibodies and antibodies against double-stranded DNA. Immuno plates were coated with 10 μg/mL purified Ig or native dsDNA antigen for detecting IgG antibodies or antibodies to dsDNA, respectively, and incubated overnight at 4 °C. After blocking with 2% FBS in 1× PBS, the plates were incubated with serially diluted standard or tested samples and then with horseradish peroxidase-conjugated detection antibodies. The absorbance was measured at 450 nm using an ELISA plate reader.

### 4.9. RNA Sequencing

Tregs were sorted from WT and AMPK-KO mice using a Facs Jazz cell sorter. The sorted Tregs were rested for 1 h in RPMI media and stimulated with 2 µg/mL anti-CD3 and anti-CD28 for 12 h at 37 °C and 5% CO_2_ for 12 h. The cells were harvested after incubation and 500 µL TRIzol Reagent (15596026, Life Technologies, Carlsbad, CA, USA) was added. The prepared samples were transferred to Ebiogen Inc. (Seoul, South Korea) and maintained at −80 °C for RNA sequencing and further experiments. The results were analyzed for the GSEA plot and heat map using ExDEGA data analysis software provided by Ebiogen Inc., Seoul, South Korea)

### 4.10. Statistical Analysis

Statistical analysis was performed using Prism 8.1 (GraphPad Software Inc.; San Diego, CA, USA). The unpaired two-tailed Student’s *t*-test was used for comparisons between two groups. Tukey’s multiple comparison test was used for multiple comparisons. Data are presented as the mean ± standard deviation (SD). Statistical significance was defined as *p* < 0.05.

## 5. Conclusions

We have revealed the role of AMPK in Tregs in decreasing inflammation during aging. Overall, our findings support the notion that AMPK in Tregs promotes the IL2–STAT5 axis for the effective suppressive function of Tregs to control inflammation during aging.

## Figures and Tables

**Figure 1 ijms-23-12384-f001:**
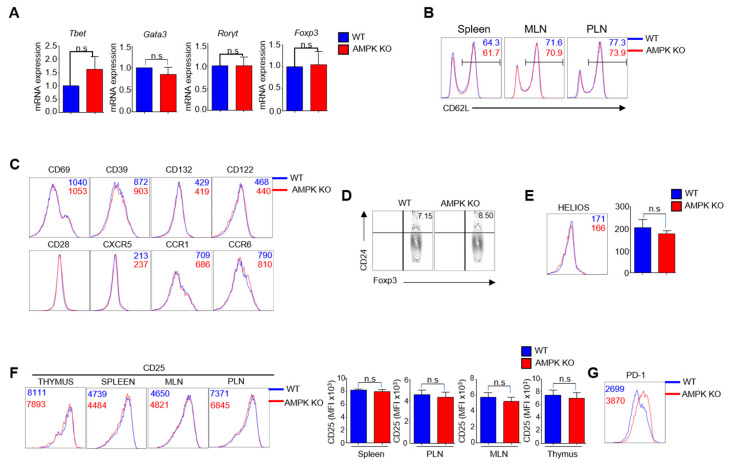
The phenotypes of young *AMPKα1^fl/fl^ Foxp3^YFP-Cre^* (AMPK-KO) and *Foxp3^YFP-Cre^* (WT) mice were similar. (**A**) Detection of mRNA expression of the *Tbet*, *Gata3*, and *Rorγt* from CD4 and *Foxp3* from regulatory T cells (Tregs) isolated from WT and AMPK-KO mice by real-time PCR. The data were normalized to *Gapdh* gene expression. (**B**) Flow cytometric analysis of CD62^hi^CD44^lo^ percentage in CD4 isolated from WT and AMPK-KO mice. (**C**) MFI representatives of CD69, CD39, CD132, CD122, CD28, CXCR5, CCR1, and CCR6 in CD4 isolated from WT and AMPK-KO mice. (**D**) The percentage of immature and mature thymic Tregs in WT and AMPK-KO mice. (**E**) Flow cytometric analysis of the MFI of the helios in Tregs from WT and AMPK-KO mice. (**F**) Flow cytometric analysis and MFI representatives of CD25 in Tregs from the thymus, mesenteric lymph nodes (MLNs), peripheral lymph nodes (PLN), and spleen from WT and AMPK-KO mice. (**G**) MFI representation of the flow cytometric analysis of PD-1 in Tregs from the spleen of WT and AMPK-KO mice. n.s means non-significant (*p* > 0.05).

**Figure 2 ijms-23-12384-f002:**
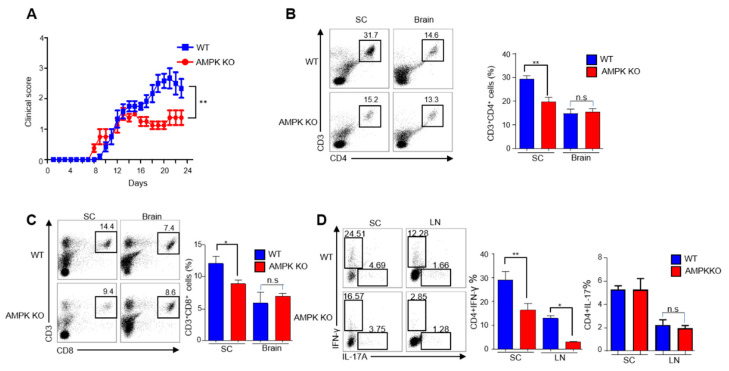
Young *AMPKα1^fl/fl^ Foxp3^YFP-Cre^* (AMPK-KO) mice show less inflammatory phenotype than *Foxp3^YFP-Cre^* (WT) mice. Active experimental autoimmune encephalomyelitis (EAE) was elicited and the clinical severity of EAE and cell infiltration in WT and AMPK-KO mice were monitored after immunizing 24–30-week-old C57BL/6J mice with myelin oligodendrocyte peptide. (**A**) Disease severity was measured in terms of clinical score based on the signs and symptoms after inducing EAE in WT and AMPK-KO mice (*n* = 5 mice per group). Flow cytometric analysis of the percentage of (**B**) CD4^+^ and (**C**) CD8^+^ T cells in total mononuclear cells obtained from spinal cord and brain of EAE-induced WT and AMPK-KO mice. (**D**) Flow cytometric analysis of the percentage of IFN-γ- and IL-17A-producing CD4^+^T cells in lymphocytes isolated from spinal cord and lymph nodes from WT and AMPK-KO mice. * *p* < 0.05, ** *p* < 0.01 and n.s means non-significant (*p* > 0.05).

**Figure 3 ijms-23-12384-f003:**
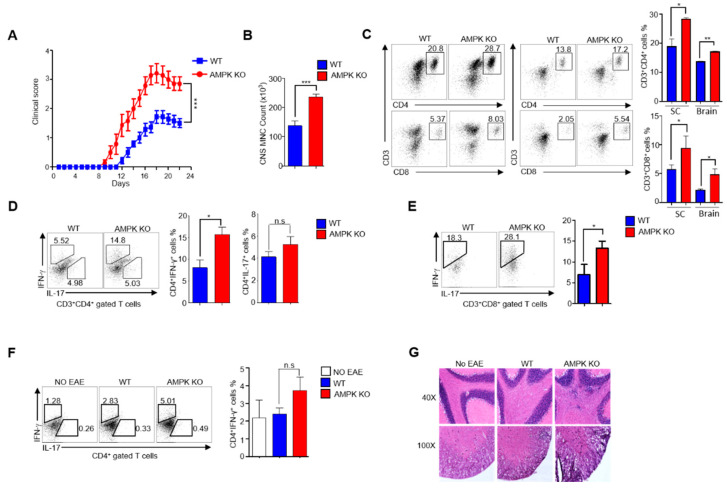
AMP-activated protein kinase (AMPK) loss abolishes the ability of regulatory T cells (Tregs) to resolve experimental autoimmune encephalomyelitis (EAE) in aged mice. Active EAE was induced in 24–30-week-old C57BL/6J mice. (**A**) Measurement of disease severity in terms of clinical score based on the signs and symptoms after inducing EAE in *Foxp3^ypf-cre^* (WT) and *AMPKα1^fl/fl^Foxp3^yfp-cre^* (AMPK-KO) mice (*n* = 5 mice per group) (**B**) Total mononuclear cell count from EAE-induced WT and AMPK-KO mice spinal cord. (**C**) Flow cytometric analysis of the percentage of CD3^+^CD4^+^ and CD3^+^CD8^+^ T cells from (left) spinal cord and (right) brain of EAE-induced WT and AMPK-KO mice. (**D**) Flow cytometric analysis of the percentage of IFN-γ and IL-17A in CD3^+^CD4^+^ gated T cells from spinal cord of EAE-induced WT and AMPK-KO mice. (**E**) Flow cytometric analysis of the percentage of IFN-γ in CD3^+^CD8^+^ gated T cells from spinal cord of EAE-induced WT and AMPK-KO mice. (**F**) Flow cytometric analysis of the percentage of IFN-γ and IL-17A in CD3^+^CD4^+^ gated T cells from the spleen of EAE-uninduced WT, EAE-induced WT, and EAE-induced AMPK-KO mice (**G**) Photomicrograph of brain sections of EAE-uninduced WT, EAE-induced WT, and EAE-induced AMPK-KO mice stained with hematoxylin and eosin (H&E) staining 22 days post EAE induction. * *p* < 0.05, ** *p* < 0.01, *** *p* < 0.001 and n.s means non-significant (*p* > 0.05).

**Figure 4 ijms-23-12384-f004:**
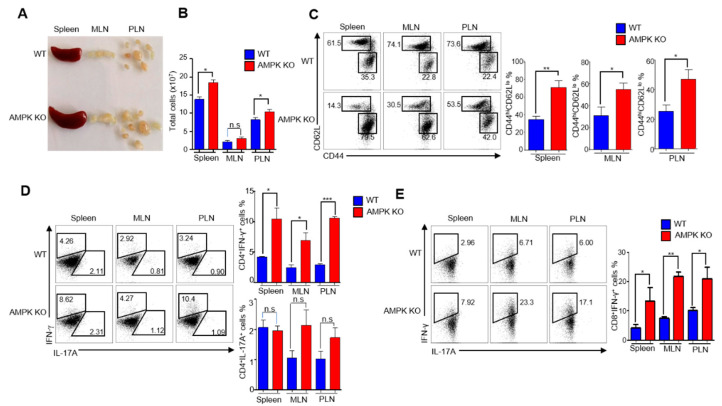
Regulatory T cell (Treg)-specific AMP-activated protein kinase (AMPK) deletion develops spontaneous systemic lymphoproliferative disease with age. (**A**) Representative image of spleen, mesenteric lymph nodes (MLN), and peripheral lymph nodes (PLN) from 24–30-week-old *Foxp3^ypf-cre^* (WT) and *AMPKα1^fl/fl^ Foxp3^YFP-Cre^* (AMPK-KO) mice. (**B**) Total cell number in spleen, MLN, and PLN of WT and AMPK-KO mice. (**C**) Expression of percentage of CD62L and CD44 in CD4^+^T cells from the spleen, PLN, and MLN from WT and AMPK-KO mice (left) and the bar diagram representatives of CD62L^hi^CD44^lo^ percentage (right). Flow cytometric analysis of the percentage of (**D**) IFN-γ- and IL-17A-producing CD4^+^T cells and (**E**) IFN-γ-producing CD8^+^T cells in lymphocytes isolated from the spleen, MLN, and PLN of WT and AMPK-KO mice. * *p* < 0.05, ** *p* < 0.01, *** *p* < 0.001 and n.s means non-significant (*p* > 0.05).

**Figure 5 ijms-23-12384-f005:**
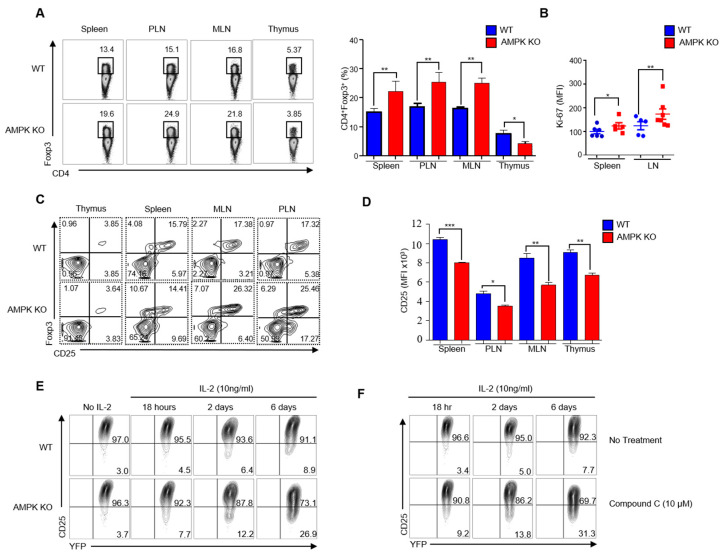
CD25 expression substantially decreased in regulatory T cells (Tregs) from *AMPKα1^fl/fl^ Foxp3^YFP-Cre^* (AMPK-KO) mice with age. (**A**) Expression of Foxp3^+^ cells gated from CD4^+^ cells isolated from the spleen, peripheral lymph nodes (PLN), mesenteric lymph nodes (MLN), and thymus of *Foxp3^ypf-cre^* (WT) and AMPK-KO mice and the frequency of CD4^+^Foxp3^+^ cells in bar diagram in those mice (right). (**B**) Scatter plot representation of MFI of Treg proliferation from the spleen and lymph nodes of WT and AMPK-KO mice according to the Ki-67 protein expression analysis. (**C**) Expression of Foxp3^+^ CD25^+^ and Foxp3^+^ CD25^−^ cells gated from CD4^+^ cells isolated from the thymus, spleen, MLN, and PLN of WT and AMPK-KO mice. (**D**) Bar diagram representation of the flow cytometric analysis of MFI of CD25+ cells gated from CD4^+^ cells isolated from the thymus, spleen, MLN, and PLN of WT and AMPK-KO mice. (**E**) Flow cytometric analysis of YFP^+^CD25^+^ cells gated from CD4^+^YFP^+^ cells after in vitro stimulation of Tregs sorted from WT and AMPK-KO mice and cultured for various time periods in the presence of only IL-2. (**F**) Flow cytometric analysis of YFP^+^CD25^+^ cells gated from CD4^+^YFP^+^ cells after in vitro IL-2 stimulation of Tregs sorted from WT mice and cultured for various time periods in the presence or absence of compound C. * *p* < 0.05, ** *p* < 0.01 and *** *p* < 0.001.

**Figure 6 ijms-23-12384-f006:**
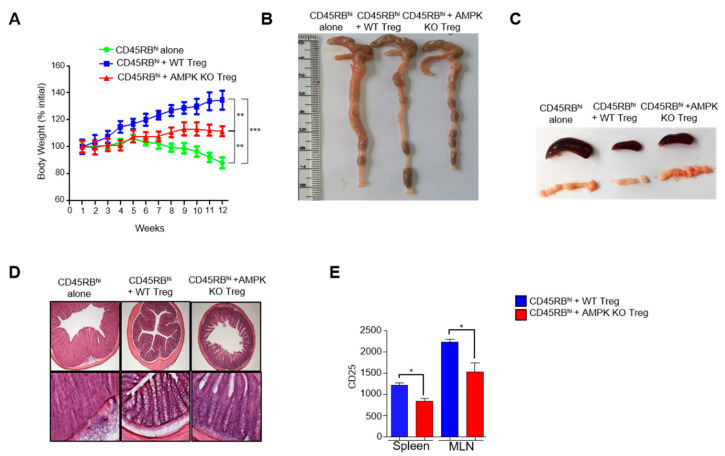
AMP-activated protein kinase (AMPK) supports suppressive activity of regulatory T cells (Tregs). Adoptive transfer of CD45RB^hi^ alone, CD45RB^hi^ + WT Tregs, and CD45RB^hi^ + KO Tregs in the disease phenotype of 6-week-old Rag^−/−^ mice and (**A**) measurement of change in body weight of the recipients (*n* = 5 mice per group). (**B**) Representative photos of the colon of each group. (**C**) Representative image of the spleen and mesenteric lymph nodes (MLN). (**D**) Photomicrograph of hematoxylin and eosin (H&E) staining of colon sections of each group; upper row 40× (Scale bar 500 μm) and lower row enlarged to 100× (Scale bar 200 μm). (**E**) Bar diagram representation of the flow cytometric analysis of MFI of CD25+ cells gated from CD4^+^ cells isolated from the spleen and MLN of CD45RB^hi^ + WT Tregs and CD45RB^hi^ + KO Tregs. * *p* < 0.05, ** *p* < 0.01, *** *p* < 0.001 and n.s means non-significant (*p* > 0.05).

**Figure 7 ijms-23-12384-f007:**
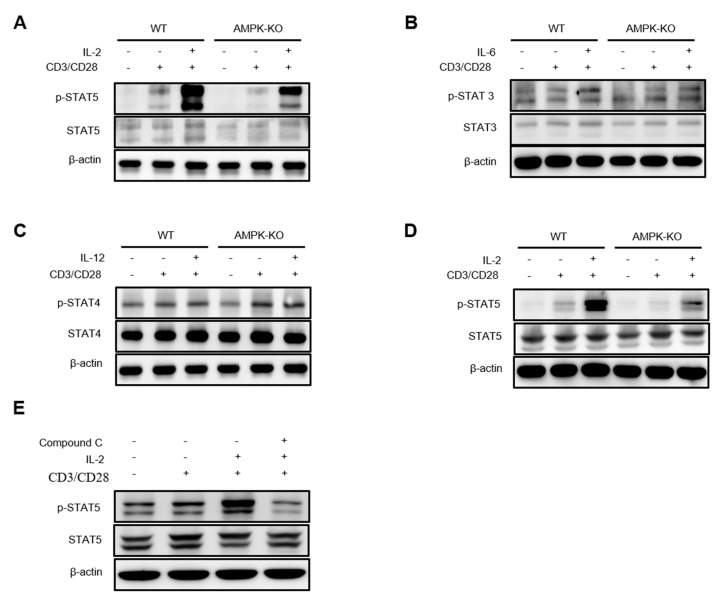
AMP-activated protein kinase (AMPK) amplifies IL-2–STAT5 signaling in regulatory T cells (Tregs). Western blot analysis of (**A**) STAT-5, (**B**) STAT-3, and (**C**) STAT-4 in WT and AMPK-KO Tregs after 1 h of resting and 12 h stimulation with anti-CD3/CD28 alone or in combination with IL-2, IL-6, and IL-12, respectively. (**D**) Western blot analysis of STAT-5 in WT and AMPK-KO Tregs from aged mice after 1 h of resting and 12 h stimulation with anti-CD3/CD28 alone or in combination with IL-2. (**E**) Western blot analysis of STAT-5 in WT Tregs after 1 h of resting and 18 h of stimulation with anti-CD3/CD28 alone or in combination with IL-2 along with treatment with compound C.

## Data Availability

Further information and requests for reagents and resources should be directed to and will be made available by the lead contact Jae-Hoon Chang (jchang@yu.ac.kr) upon reasonable request.

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
