# Peer review of "AMPK Amplifies IL2–STAT5 Signaling to Maintain Stability of Regulatory T Cells in Aged Mice"

_ijms, 2022, doi:10.3390/ijms232012384_

Round 1

Reviewer 1 Report

The authors investigated the role of AMPK in the stability of Treg and inflammation of young and aged mice. In young mice, the loss of AMPK leads to a less severe EAE while aged AMPK-KO mice fail to control inflammation through the IL-2-STAT5 axis. 

The manuscript is well written. The results shown by the authors are enough convincing and the title is representative of the manuscript content. The authors highlighted the novelty of this manuscript and described a future approach such as exploring the mechanism of AMPK-mediated STAT5 phosphorylation. 

In my opinion, the blot membranes should let us see a larger protein standard range to confirm the protein's molecular weight. 

Minor changes:

- Lines 13-15: the sentence is confusing. Clarify, please. 

Author Response

The authors investigated the role of AMPK in the stability of Treg and inflammation of young and aged mice. In young mice, the loss of AMPK leads to a less severe EAE while aged AMPK-KO mice fail to control inflammation through the IL-2-STAT5 axis. 

The manuscript is well written. The results shown by the authors are enough convincing and the title is representative of the manuscript content. The authors highlighted the novelty of this manuscript and described a future approach such as exploring the mechanism of AMPK-mediated STAT5 phosphorylation. 

In my opinion, the blot membranes should let us see a larger protein standard range to confirm the protein's molecular weight. 

Thank you for your comment. We cut the membranes on the basis of their molecular weight with reference to protein standard ladder to observe the desired proteins during exposure. Also, we have sent the original blot membranes while submitting our manuscript.

Lines 13-15: the sentence is confusing. Clarify, please

Thank you for your comment. In line 13-15, we wanted to conclude that, our study addressed the role of AMPK in Tregs in sensing IL-2 signaling to amplify STAT5 phosphorylation which support Treg stability by maintenance of CD25 expression thereby controlling inflamm-aging.

Reviewer 2 Report

The work entitled "AMPK amplifies IL2–STAT5 signaling to maintain the stability of 

regulatory T cells in aged mice" done by Pokhrel et.al. focused on elucidating the role of AMPK, a serine/threonine protein kinase in the functioning of the Tregs by using a mouse KO system and tested it using different disease models. The role was more focused on the stability and function of the Tregs population to control inflammation in aging mice. 

The authors have a clear hypothesis and rationale for the work and presented it by conducting a streamlined and logical experimental approach first by testing the expression and function of the Tregs derived from the AMPK KO mice and showing that there is no phenotypic difference between the control and the KO mice Tregs by checking the expression of surface markers. They also found that there is no difference in the developmental stages of T-cells. They found that AMPK KO T-cells from young mice have a less inflammatory phenotype compared to the age-matched wild-type mice. Later they used EAE as a model to show that the loss of AMPK in Tregs is associated with its inability to resolve the EAE in aged mice as seen by the loss of controlling the inflammatory cytokine production. This is also seen in the immune infiltration in the brain sections and the increased IFN gamma-producing cells. These AMPK KO mice were further shown to develop spontaneous systemic lymphoproliferative disease with age. This was associated with the decrease in the expression of CD25 using RNA seq and also shown to have an increase in the CD25-FoxP3+ cells in the AMPK KO mice. The authors showed that AMPK is essential for the activity of Tregs by inducing colitis using adoptive T-cell transfer. Finally, the authors showed the possible mechanism by which AMPK regulates the Tregs status by showing the activation of IL-2-STAT5 signaling.

The quality of data presented and the statistical test are done appropriately to the best of my knowledge. The materials and methods section is complete and gives all the necessary information.  Finally, the discussion is well written and discusses the possible mechanism and also give a drawback to the study, and proposes further studies. However, there are some concerns 

1.     In figure 1 the authors should also check the expression of CD40L and may wish to cite PMID: 32827854 and 32827854 it for reference since CD40L is a curtail co-stimulation receptor on T-cells and their status has to be checked. Also, the regulation of  STAT3  signaling.

2.     In figure 1 the authors should also show the IFN gamma and IL-7 populations from the brain fractions along with the spinal cord and the lymph node. 

3.     The authors should also consider including the transcription factors for the IL-17A and IFN gamma-producing T-cells as they are important. Similarly, it should be shown in figure 3

4.     The authors should show the statistical significance stars where ever it is significant and mention n.s. if it is not.

5.     In figure 3 which shows the immune infiltration in the brain, the authors should show a quantitative measure for the staining. 

6.     The authors should show the saline control and show the statistical significance between the different groups. The adoptive transfer of CFSE labeled T-cells will give a much clear understanding of whether the T-cells are proliferating and are in the mice at end of the 13 weeks study. Also, the authors should show the cytokine profile of the T-cells if these are associated with decreased inflammation. 

7.     Line 170 and 188 shows CD62 instead of CD62L.

8.     The horizontal line in the 5D figure is tilted.

Overall the work done by Pokhrel et.al is commendable and adds to the available information on the loss of Trges ability upon aging.
